# Characterization and under Water Action Behaviour of a New Plaster-Based Lightened Composites for Precast

**DOI:** 10.3390/ma16020872

**Published:** 2023-01-16

**Authors:** Manuel Álvarez, Daniel Ferrández, Patricia Guijarro-Miragaya, Carlos Morón

**Affiliations:** Departamento de Tecnología de la Edificación, Escuela Técnica Superior de Edificación, Universidad Politécnica de Madrid, 28040 Madrid, Spain

**Keywords:** super absorbent polymer, water performance, building precast, geomaterials

## Abstract

Plaster is a construction material widely used for the production of prefabricated parts in building construction due to its high capacity for hygrothermal regulation, its good mechanical performance, and its fireproof nature, among other factors. Its historical use has been linked to ornamental elements, although more recent research is oriented towards the industrialisation of plaster composites and the design of prefabricated parts for false ceilings and interior partitions. In this work, the behaviour against water of four new plaster-based composite materials is studied, using additions of two types of super absorbent polymers (sodium polyacrylate and potassium polyacrylate) and a lightening material (vermiculite) in their manufacturing process. In addition, the transmission of water vapour through the samples was studied together with the water absorption capacity of the samples in order to check the suitability of the use of plaster-based materials exposed to these environments. The results of this study show that composites with the addition of super absorbent polymers as well as vermiculite significantly improve their water performance compared to traditional materials up to 7.3% water absorption with a minimal (13%) reduction in mechanical strength compared to current materials with similar additions. In this sense, a plaster material is obtained with wide possibilities of application in the construction sector that favours the development of sustainable and quality buildings, in line with Goal 9 for Sustainable Development included in the 2030 Agenda.

## 1. Introduction

Plaster is still today one of the most widely used building materials in construction [1]. Among the main advantages of this material are its versatility and ease of application. Plaster powder mixed with water generates a liquid paste that allows any type of shape to be formed and can be applied directly in the execution of surface finishes or for the production of prefabricated elements [2]. In this sense, as a result of its purity and fineness of grind, plaster becomes an optimal material for interior finishes in buildings [3].

Despite its great advantages, exposure to the action of water remains one of the major limitations of gypsum composite materials for use in the building sector [4,5]. The behaviour of these materials when in contact with water is still a health problem [6]. As this is a disadvantage of these materials, most of the studies in the existing literature do not carry out water resistance tests or trials, focusing the research on the mechanical, thermal, and acoustic characterisation aspects of the developed composites [7,8,9,10,11,12]. Although it is true that, on some occasions, the coefficient of water absorption by capillarity is determined, this information is usually treated in a complementary manner without going into the problem in depth [9]. In this sense, as water behaviour is one of the main weaknesses of gypsum composites, it becomes a great opportunity to carry out studies to understand and characterise these properties with the intention of introducing improvements in the manufacturing process of these materials for their use in wet chambers.

Among the possible solutions is the incorporation of plastic materials into the manufacturing process of gypsum and plaster composites. These polymeric materials are positioned as a technically valid and economically viable option to improve the performance of prefabricated plaster and plasterboard commonly used in buildings [13,14]. In this sense, some studies show how the incorporation of plastic waste in the plaster composite matrix improves the performance of these materials, reducing the total water absorption coefficient, and decreases their permeability to water vapour [15]. These additions also make it possible to introduce circular economy criteria into the building process, favouring waste management and reincorporating construction and demolition waste into the production system of new prefabricated products [16]. In addition, it has been possible to verify how plaster composites with the incorporation of polymeric materials in their composition make it possible to obtain mechanical strengths that meet the requirements of the UNE-EN 13279-2 standard for gypsum composite materials applied in construction [17,18], as well as to achieve a higher acoustic absorption coefficient compared to traditional plaster [19,20].

Another of the most studied polymers for addition to plaster composites is polyvinyl acetate (PVA). Used in conjunction with polypropylene fibre reinforcement (0–1.2% volume fraction), Zhu et al. developed a new composite with a faster curing process and improved both the flexural strength and toughness of the material by about 20% [21]. In other research, polymeric composites have been used to modify the matrix of plaster composites, favouring the occurrence of chemical reactions during the setting process that allow the introduction of occluded air into the material, thus reducing the density of the final composite [22]. 

Expanded polystyrene (EPS) and extruded polystyrene (XPS) residues represent some of the most widely used polymeric materials to reduce the density of gypsum and plaster composites, with a wide field of application in the production of prefabricated slabs and panels with low thermal conductivity and good acoustic behaviour, as reported in the literature [5,11,23]. Among the most recent research is that conducted by Bicer et al., who developed an experimental campaign in which, by adding small-diameter EPS and tragacanth, they managed to improve thermal conductivity by 20% and reduce water absorption by 30%, making this type of polymer a valid resource for improving comfort conditions in homes [24].

Finally, there are those known as super absorbent polymers (SAPs). These are polymers supplied in granular form that are capable of absorbing and retaining large amounts of water. These materials are highly resistant to attack by large chemical agents such as acids or chlorides [25,26]. They are mainly used in agricultural sectors so that they can be mixed with the soil to store water [27]. In building construction, they are often used as a sealant in underwater concrete joints [28], as well as sheathing for conductor cables to ensure their proper functioning. A study by Liu et al. showed another possible use for these materials, as when applied to cold-formed metal structures, they prevent fires by increasing their fire resistance [29]. However, despite these advantages, these polymers are among the least used for application in construction materials. 

The main objective of this research is to analyse the behaviour against water of a new plaster composite material made with SAPs and lightened with vermiculite. Therefore, an experimental campaign is proposed in which different types of plaster with additions of sodium polyacrylate and potassium polyacrylate with and without the incorporation of vermiculite are produced, analysing their physical–mechanical properties, and studying the technical feasibility of this new material for use in prefabricated panels and slabs. 

## 2. Materials and Methods

### 2.1. Materials

Figure 1 shows the raw materials used to develop the proposed research: plaster, water, sodium polyacrylate, potassium polyacrylate, and vermiculite.

#### 2.1.1. Binder

Iberyola E-35 fast-setting plaster, supplied by Placo Saint-Gobain in accordance with the UNE-EN-13279-1 standard [30], was used as the matrix element. Its main characteristics include its thermal conductivity of 300 mW/m-K, its purity index of over 90%, and its fineness of grind, with a particle size of 0–0.2 mm [31]. In addition, this binder is classified as A1 in term of reaction to fire according to Spanish regulations [32].

#### 2.1.2. Water

The water used for this study was tap water from Canal de Isabel II (Madrid, Spain). The chemical analysis conducted annually by the water service of the Madrid City Council accredits the absence of harmful chemical agents such as chlorides, sulphates, etc., in sufficient quantities to alter the properties of the plaster paste. The result is an inert water, of a soft nature, and classified as acceptable for human consumption [33].

#### 2.1.3. Super Absorbent Polymers (SAPs)

Table 1 shows the main chemical characteristics of the two super absorbent polymers used in this research [26,34]. The incorporation of these polymers into the manufacturing process of the plaster composites allows for the generation of an internal pore network in the matrix of the hardened material. This effect is due to the evaporation of the water absorbed by these SAPs during the setting process of the plaster composites. Additionally, this evaporation is favoured by the oven drying process of the hardened samples after seven days according to UNE-EN-13279-2 [35].

#### 2.1.4. Vermiculite

Vermiculite is a naturally occurring material of volcanic origin consisting of iron or magnesium silicates, belonging to the mica group [23], whose chemical formula (Formula (1)) is as follows:(1)$$\left(\mathrm{Mg},\mathrm{Ca}\right)\;0.7\;\left(\mathrm{Mg},\mathrm{Fe},\mathrm{Al}\right)\;6.0\;\left[({\left(\mathrm{Al},\mathrm{Si}\right)}_{8}O_{20}\right)]\;\left(\mathrm{OH}\right)\;4.8{\;H}_{2}$$

The sheets that form the vermiculite reflect and disperse the energy it receives, resulting in good thermal insulation performance. In addition, these omnidirectional sheets reflect sound waves, while another part of them is absorbed by the air inside. This makes vermiculite also a good thermal insulator at many frequencies. As a volcanic material, its melting point is quite high at 1370 °C, making it a non-combustible and very stable mineral. This makes it a suitable material for fire retardation and fire insulation. When subjected to high temperatures, it expands, resulting in a lightweight, non-flammable material with a high capacity to absorb and retain water. Table 2 lists the physical and chemical characteristics of this product.

### 2.2. Dosages and Samples Elaboration

For this study, two SAPs and a lightening compound were used, so a total of five dosages was prepared the manufacturing process was manual and in accordance with the current UNE-EN-13279-2:2014 standard [35]. Once demoulded, they were stored in the laboratory for one week at constant temperature and relative humidity (RH) (23 °C ± 2 °C and 50% ± 5%). Subsequently, the specimens were dried after seven days for 24 h prior to testing using a LabProcess [36] oven, model ARGO LAB TCN200, at a temperature of 45 °C ± 1 °C.

Table 3 shows the dosages and composition used. The nomenclature is as follows: E0.7-SAP-V, where E0.7 refers to the water/plaster ratio by mass used, SAP indicates the type of polymer added, which can be sodium polyacrylate (Na) or potassium polyacrylate (K), and finally V refers to those compounds that add the vermiculite lightening load.

The amounts of SAP and vermiculite were incorporated as additions to the plaster paste, without there being any substitution of the raw materials that make up the matrix of the composite. In the same way, the lightening agents were added to the matrix as an admixture. In total, 15 g of SAP and 30 g of vermiculite were added.

To develop this study, tests described below were conducted. With this, 7 series of three specimens of each chosen dosage were prepared. Table 4 shows the dimensions and the tests conducted on each of these specimens.

### 2.3. Experimental Plan 

As the aim was to check composites’ water reaction studied, a series of tests, some of which are not standardised, was conducted based on the existing literature for characterising gypsum and plaster composites [37,38,39,40]. All tests were conducted in the construction materials laboratory of the Escuela Técnica Superior de Edificación de Madrid. As superabsorbent polymers were used, which contribute to the water absorption of the materials used, the mechanical characterisation was meaningful, as well as the reaction to exposure of the different tests to which these materials were exposed. 

#### 2.3.1. Water Absorption

The aim of these tests was to characterise the water absorption of the materials. To this end, three different tests were conducted. On the one hand, the water retention test included in the annex of the UNE-EN-13279-2 standard [35], which expresses water retention as a percentage of the total mass, was conducted. On the other hand, the non-standardised capillary absorption test was conducted to determine the height that the rising water reaches through the plaster material, and finally, the total water absorption coefficient was determined.

The capillary water absorption test lasts 15 min and is conducted as follows. A container is filled with water to a height of one centimetre. A grid-like support is then placed at a height of one centimetre, so that the water is flush with the support. Once this is done, the 4 × 4 × 16 cm^3^ samples are placed vertically, and the stopwatch is started. After the first five minutes, a line is drawn at the height the water has reached on one side. After this first measurement, a new line is drawn every minute. At the end of the test, the height in mm reached each minute is measured and recorded, and the result is expressed in mm/min.

On the other hand, for the water retention test, the materials needed for this test are the following: filter paper, non-woven gauze, Ø140 mm circular plastic ring, two square plastic plates at least 30 mm larger than the circular ring, and the mortar in question. The purpose of this test is to measure, during the setting process, the amount of water retained by the mass, by means of the difference in weights. The complete development is set out in the UNE-EN-459-2:2021 standard [41]. 

Finally, total water absorption was conducted. The purpose of this test is to check the total water absorption capacity of the material, its capacity to lose water, and how it affects its mechanical behaviour. This test is determined in the UNE-EN-520 standard [42]. It consists of completely submerging the samples for two hours in a container with water. Once this has been done, they are weighed to obtain the total amount of water absorbed. They are then stored in a laboratory environment (21 °C and 35% RH) for seven days. They are then weighed again to check the amount of water they have been able to remove. Once this has been done, they are mechanically characterised by testing surface hardness, flexural strength, and compressive strength.

#### 2.3.2. Water Vapour Permeability

Water vapour permeability is another of the variables studied in this article. This test is included in the UNE-EN-ISO-12572 standard [43]. For this purpose, samples of each dosage were prepared with a thickness of 1 cm and a diameter of 16.5 cm, as shown in Table 4. These samples were placed in an airtight plastic container containing 200 mL of aqueous solution. To guarantee an environment of 98% relative humidity inside the containers, the dissolution was conducted until the water was saturated by the addition of potassium nitrate salts. In addition, to ensure that the water vapour diffused only through the plate, the edges of the plate were sealed to the vessel with silicone. 

After the preparation process, twice a week for eight weeks, the different samples were weighed with the container to record the water vapour losses. With these weights, and using the procedure indicated in the standard, the water vapour resistance “R” was obtained.

#### 2.3.3. Wet Chamber

The purpose of this test is to observe the variations in weight and mechanical behaviour of the dosages studied when subjected to a constant temperature and high relative humidity for a prolonged period of time. This test is not standardised but follows the test conducted by del Río Merino in his doctoral thesis [40]. It consists of reserving the samples for five days in a wet chamber with hygrothermal conditions of 21 °C and 75% relative humidity. 

After this time, the specimens were weighed to calculate the percentage of water acquired during this time. The mechanical characterisation of the specimens was then conducted, consisting of shore C surface hardness, flexural strength, and compressive strength tests. 

#### 2.3.4. Water–Stoves Cycles

The purpose of the water–stove cycling test is to test the response to exposure to extreme contrasts of humidity and dryness. After the test specimens have been made and left in a laboratory environment for seven days (23 °C and 35% humidity), they are placed in a container with water so that they are completely submerged for two days. After this time, they are removed, weighed, and placed in an oven at 60 °C for a further two days. The test tubes are weighed again, and this cycle is repeated twice. After finishing, the mechanical characterisation is conducted, consisting of measuring the surface hardness and the flexural and compressive strengths. This test is not standardised; as in the previous test, the test conducted by del Río Merino in his doctoral thesis [40] was followed. 

#### 2.3.5. Mechanical Characterisation

With the exception of the water vapour permeability, capillary absorption, and water retention samples, all samples produced were tested for flexural and compressive strength and Shore C surface hardness in order to compare them with the reference samples as shown in Figure 2.
Bending and compressive test. These tests were conducted according to UNE-EN-13279-2 [35] using a hydraulic press machine, model AUTOTEST 200-10SW, from IBERTEST (Madrid, Spain).Surface hardness. It was conducted with the help of a Shore C durometer.

## 3. Results and Discussion

This section presents the results derived from the planned experimental campaign, as well as the discussion derived from their interpretation.

### 3.1. Water Retention and Absorption and Capillarity Test

This section shows the results obtained for the three tests related to water absorption in the plaster composites produced. Firstly, in Table 5, the values obtained for the water retention test are shown. 

The water retention test shown in Table 5 was used to measure the amount of water that the plaster mixture was able to release when setting. The results of this test showed a water retention of 61.77% for the E0.7 dosage. This means that it lost a considerable amount of water compared to that initially existing in the mixture. Those dosages with the addition of sodium (80.33%) and potassium (84.2%) polyacrylates showed higher water retention due to the nature of these superabsorbent polymers. Finally, it was observed that the incorporation of vermiculite in the preparation of the plaster composites reduced the amount of water retained in the samples prepared with SAP, although without reaching the values obtained for the reference plaster E0.7.

Figure 3 shows the results obtained for the total water absorption test and the height reached after the capillary water absorption test.

The capillary water absorption and total water absorption tests analyse material behaviour once it has hardened. Therefore, they are related to material response throughout its service life. Figure 3 shows how the dosage without any addition (E0.7) reached the highest height of all (56 mm) and retained the highest amount of water when submerged (38.8% of its initial weight). The addition of SAP to the mixture reduced the total absorption of both by 4.1% for sodium polyacrylate and 4.6% for potassium polyacrylate. Likewise, the capillary height also decreased in both cases, being lower in the samples with the addition of potassium polyacrylate, as with the total water absorption. This effect was more pronounced in the plaster composites incorporating vermiculite in their composition compared to the E0.7 reference, further reducing the values obtained in the samples containing only SAP. 

### 3.2. Water Vapour Permeability

Figure 4 shows the relative water vapour permeability results. It can be seen that the behaviour of most of the compounds was quite similar. From week one to two, there was a high diffusion of water vapour through the samples. From week two to five, the value remained stable with small variations. Finally, from week five to eight, water vapour permeability decreased 50%. 

Although all the plaster composites studied followed the same pattern, the E0.7-Na sample presented a greater opposition to the diffusion of water vapour through its surface, obtaining lower transmission values than the rest, especially in the central weeks (2–5), where the difference was up to 25%. Week seven was the most stable and similar for all samples, as the maximum difference in transmission was 2%. It should be noted that all samples with polymer addition reduced the water vapour permeability compared to the reference. This is mainly due to the water absorption capacity of the polymers, which trapped water vapour while passing through the samples. This resulted in a clogging of the pores generated in the plaster matrix, thus reducing the surface area where water vapour could pass through. 

In this test, in addition to determining the permeability to water vapour passing through the processed samples, the water vapour retention capacity inside each of the plaster composites could be observed. Table 6 shows the mass variation observed in the samples before and after completion of the water vapour permeability test, as well as the percentage difference between the two. As a first result, it should be noted that all the samples tested retained water. The sample E0.7-Na-V was the one that retained the highest amount of water, reaching 1%, which was twice the amount obtained in samples with the lowest amount of water retention, namely, E0.7-K and E0.7 (0.5%). Samples with the addition of vermiculite increased the amount of water retained after the test. Although not very significantly, this may be due to the greater specific surface area in these plaster matrices as a result of the expanded vermiculite. This allowed water vapour to condense on its surface when it passed through the material.

### 3.3. Wet Chamber

The wet chamber tests (21 °C and 75% relative humidity) yielded two different types of results. On the one hand, the percentage of water absorption by the specimens during the duration of the test, which can be observed in Table 7, was determined. On the other hand, the results derived from the mechanical characterisation, hardness, bending, and compression are shown in Section 3.5 and were compared and analysed with the other tests conducted in the mechanical characterisation section. 

All tested samples retained a higher percentage of water compared to the reference plaster E0.7 (1.15%). However, contrary to the water vapour transmission test, those samples with added vermiculite retained less water than those containing only SAP. The uninterrupted exposure of the specimens to the wet environment for two days showed a higher water retention than water vapour transmission samples even though they were subjected to a saturated environment for a longer period of time. 

### 3.4. Water–Stove Cycles

The results of the water–stove cycle test was analysed by comparing the weight increase at the end of each of the cycle phases (first cycle C1 and second cycle C2). Table 8 shows, in grams and percentage ratio, the differences obtained.

Water absorption by the samples analysed was lower in the E0.7 reference (32%) than in all samples with additions. In water–stove cycles, samples with the highest percentage of water retention were those without vermiculite, namely, E0.7-Na and E0.7-K, with 38–39% and 38–38% in the respective cycles 1 and 2. The addition of vermiculite resulted in a 2–3% difference compared to samples with only SAP. It should be noted that after the drying cycles, the weight achieved was in all cases lower than the initial dry weight prior to the test. The difference was higher than 1% in all samples.

### 3.5. Mechanical Characterization

In this section, the mechanical characterisation results between the reference, the water and stove cycles, and the wet chamber are analysed and compared. 

Figure 5 shows the results of the surface hardness tests in Shore C units. It can be observed that the results of all the tests were lower than the reference ones regardless of the type of sample tested. The maximum hardness value reached was 89 Shore C in the reference sample E.7-K-V, while the lowest value was obtained in the reference sample E0.7 subjected to water and stove cycles, with 73 Shore C. 

Both in the reference samples and in the samples subjected to durability cycles, all the mixes with the addition of SAP and vermiculite exceeded the surface hardness values obtained by the reference plaster E0.7. This translates into a direct improvement of this material characteristic. The samples with the addition of potassium polyacrylate achieved the most extreme results for these tests in such a way that the E0.7-K-V mix subjected to water–stove durability cycles obtained a reduction in its hardness with a maximum value of 11.2% compared to reference, and the E0.7-K sample subjected to cycles reduced its surface hardness minimally (5.9% compared to reference). 

The results derived from the flexural and compressive strength tests of the different plasters produced are shown in Figure 6. 

As can be seen, none of the samples with SAP and vermiculite addition showed flexural strength values higher than sample E0.7. Furthermore, in all the plaster composites studied, a decrease in flexural strength was observed when the samples were subjected to ageing cycles under the action of water. On the other hand, the wet chamber cycles were less aggressive than the water–stove cycles, since the latter had a greater effect on the mechanical behaviour of the lightened plaster composites produced in this research.

Figure 6 shows how all the plaster composites with potassium polyacrylate incorporation showed a higher flexural strength than those made with sodium polyacrylate addition. Likewise, in all the samples with the addition of SAP in which vermiculite was incorporated, there was a decrease in their mechanical strength. In all the mixes tested, flexural strength values higher than the minimum recommended in the UNE-EN-13279-2 standard [35] were obtained.

Finally, Figure 7 presents the compressive strength results of the samples tested. As in the flexural tests, no sample exceeded the strength obtained for reference E0.7 plaster (10.8 MPa). The lowest strength values were obtained by the E0.7-Na-V sample (5.9 MPa) subjected to water–stove cycles. Again, the samples tested using wet chamber cycles performed better than those exposed to water–stove cycles. This is because the changes produced inside the specimens in the water–stove cycles caused a weakening of the gypsum crystals, which had a direct impact on their internal bonding and thus on their mechanical strength [44].

In order to better compare the results and clearly establish the conclusions of the present study, the results obtained in the different tests are shown together in Figure 8. In addition to the water performance analysed, the average bulk density of the processed samples has been added. The figure also shows the minimum values required by the standards for the mechanical resistance to bending (1 MPa) and compression (2 MPa).

A direct relationship between the incorporation of SAP and vermiculite and a significant decrease in density can be observed in Figure 8, as occurred in other studies in the existing literature [44,45,46]. Compared to reference sample E0.7, the lightened plaster composites with SAP addition showed an average decrease in bulk density of 2–3%, and the samples with vermiculite addition further reduced the density to obtain a 6–8% lightened composite compared to traditional plasters. The density of the samples with potassium polyacrylate was lower than those with sodium polyacrylate, mainly due to the size and physicochemical characteristics of these additions. Potassium polyacrylate has a greater facility to release cations than sodium polyacrylate. Thus, interacting with water makes it more hydrophilic. In water behaviour tests, it was observed that in samples containing polyacrylates there was a loss of mass compared to the reference, which has a direct impact on bulk density. This difference was between 3% lower in those subjected to a wet chamber and 6% in those subjected to water and stove cycles. There was also a clear relationship between density and mechanical strength. These results obtained showed a whole new way to study these kinds of materials, as there are no similar studies in the literature. The main use of plaster-based materials for interior uses has led to a lack of standardized tests to study its behaviour under water action. 

Generally, results of the different tests were similar. The mechanical strength values (bending and compression) achieved by the samples subjected to the wet chamber test showed that the exposure of this material to wet areas did not change its mechanical strength significantly. This means that composites with SAP behaviour are acceptable for use in the manufacture of precast products in wet environments, and the values obtained were well above the minimum required by the standards [31]. Water absorption by the material in these environments helps to regulate the relative humidity in damp rooms.

The flexural strength decreased with specimens subjected to the wet chamber and decreased even more for those specimens subjected to water–stove cycles. The flexural strength of the tested specimens ranged from 2.5 MPa (E0.7-K-V subjected to water–stove cycles) to 4.1 MPa (E0.7 subjected to the wet chamber). Despite this decrease, the minimum value, E0.7-K-V subjected to water–stove cycles (2.5 MPa), was only 35% lower than E0.7 subjected to the same test and 52% lower than the reference sample E0.7 (5.2 MPa). However, this value was still above the 2 MPa required by the standard [35].

The same situation occurs with compressive strength. The decrease was greater in lightened plaster composites subjected to water–stove cycles than in those subjected to the wet chamber. After the accelerated ageing cycles, the results ranged from 6.3 MPa (E-0.7-K-V subjected to water and oven cycles) to 9.5 MPa (E0.7 subjected to the wet chamber test). As can be seen, E0.7-K-V was once again the sample with the worst results, with a decrease of 41% compared to reference E0.7 and 33% compared to E0.7 subjected to the same test. 

Despite of the good results obtained and behaviour under water action performance of this new material, there were some shortcomings that could not be avoided and considered. The non-standardized tests performed were based in previous works that are hardly endorsed. Thus, they are still non-standardized, which means that there may be some variables that were not considered and could be improved.

Finally, Figure 9 presents all results together to establish a selection of the samples with the best and worst behaviour under water action. In all the lightened plaster composites, a linear decrease in their mechanical properties was observed when subjected to accelerated water exposure cycles. It was observed that water action reduced the technical performance of the plaster materials designed for this research. It was observed that the lightened plaster composite with the best mechanical behaviour was 0.7-K subjected to wet chamber cycles, while the sample with the worst mechanical behaviour after the accelerated water exposure cycle tests was E0.7-Na-V. This combination was the one that offered the lowest guarantees for the production of prefabricated elements exposed to wet areas. The samples subjected to wet chamber testing showed more homogeneous bending behaviour than those subjected to water–stove cycles. There was a significant difference in behaviour between the reference samples and those subjected to water behaviour tests. While the reference samples progressively and linearly reduced their strength, the samples subjected to water reaction tests moved away from this distribution. In both tests, E0.7-Na and E-0.7-K-V obtained more similar values in compressive tests and E0.7-Na-V and E0.7-K in bending tests. Finally, it can be seen that the durability test water–stove cycles generated more harmful effects and a decrease in mechanical resistance compared to the wet chamber cycle test. In general, samples with the same composition performed the same way in mechanical strength tests after being subjected to accelerated water exposure cycles.

## 4. Conclusions

In this research, the water behaviour of a new type of lightened plaster composite was studied. These plaster composites have been registered under invention patent ES-2841130-B2 [47] and were specially designed for the production of prefabricated products. The results obtained in this study allow us to study their technical feasibility for use in the manufacture of slabs and panels. This also allows us to explore their application possibilities for the interior cladding of wet areas in both residential and service buildings. Finally, after reviewing and discussing the results, the following conclusions were reached.
Firstly, related to these materials’ physical properties, bulk density decreased as both SAP and vermiculite were added. Density was lower in samples with potassium polyacrylate addition than those with sodium polyacrylate addition. This density reduction resulted in lower building loads, as well as lower transport costs and ease of on-site execution.Water–stove cycle samples obtained a lower mechanical strength than wet chamber cycle samples. However, after both tests, the samples continued showing mechanical strength results suitable for development and exposure to wet environments.Wet environments exposure and water action on these materials have repercussions for their mechanical behaviour, although they do not prevent or limit their use in wet areas in buildings. All lightened plaster composites produced in this work exceeded the minimum flexural and compressive strength values set out in the UNE-EN 13279-2 standard. The worst mechanical behaviour in all the tests conducted was shown by the E0.7-Na-V composite.The lowest water vapour permeability was reached by E0.7-Na, with 27% less than the rest of the samples. However, all the lightened composites acted in a similar way to each other, improving the lightened plaster composites’ hygroscopic regulation capacity. The addition of sodium and potassium polyacrylate increased plaster composites’ water absorption capacity and reduced the capillary height reached by the water.

These conclusions have a number of implications for the application of these composites in the building sector. The good results obtained allow us to affirm that SAPs are a great opportunity for the development of prefabricated materials exposed to water action and wet environments. Materials with this addition maintain a mechanical behaviour well above the minimum required for this performance. As a main consequence, they could improve workability, as well as help to stabilise the hygrothermal conditions of the places where they are exposed. With this, pathologies resulting from condensation, both superficial and interstitial, would be avoided. Finally, the main limitation is the higher cost of the final lightened plaster product and the decrease in its mechanical properties.

## Figures and Tables

**Figure 1 materials-16-00872-f001:**
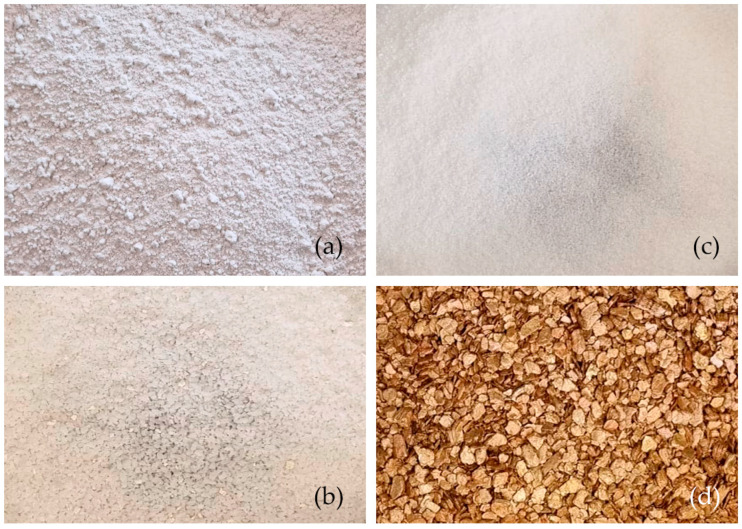
Materials used: (**a**) gypsum plaster; (**b**) potassium polyacrylate; (**c**) sodium polyacrylate; (**d**) vermiculite.

**Figure 2 materials-16-00872-f002:**
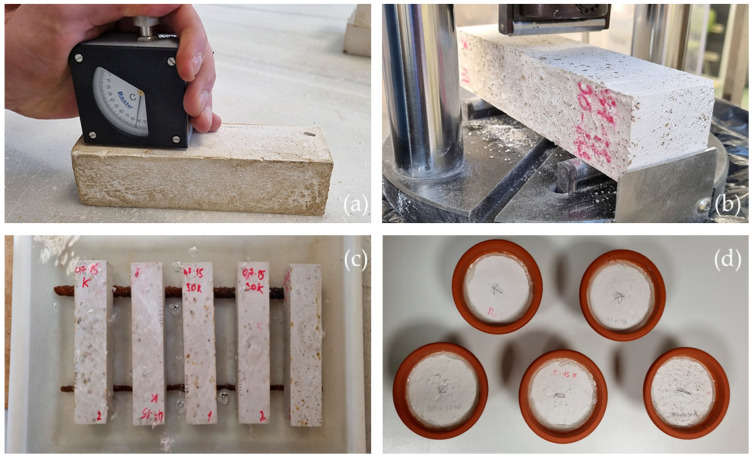
Tests performed: (**a**) surface hardness; (**b**) flexural and compressive strength; (**c**) total water absorption; (**d**) water vapour permeability.

**Figure 3 materials-16-00872-f003:**
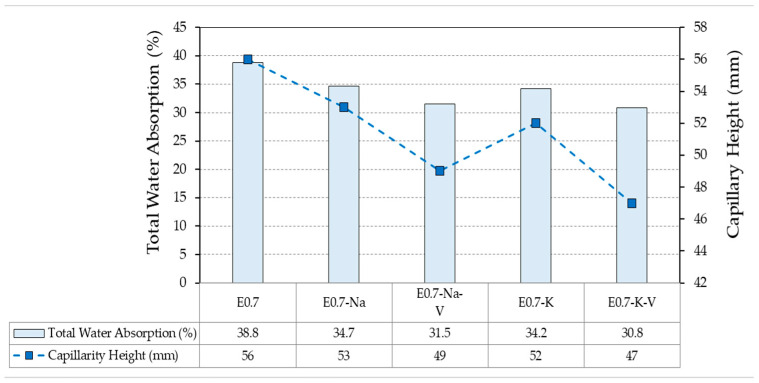
Capillarity height and total water absorption test results compared.

**Figure 4 materials-16-00872-f004:**
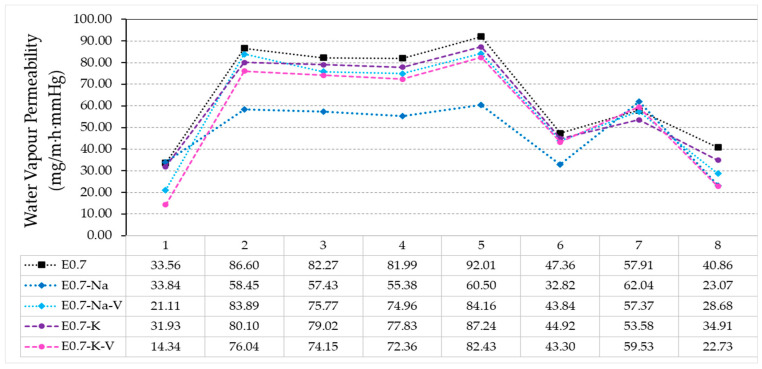
Relative water vapor permeability test results.

**Figure 5 materials-16-00872-f005:**
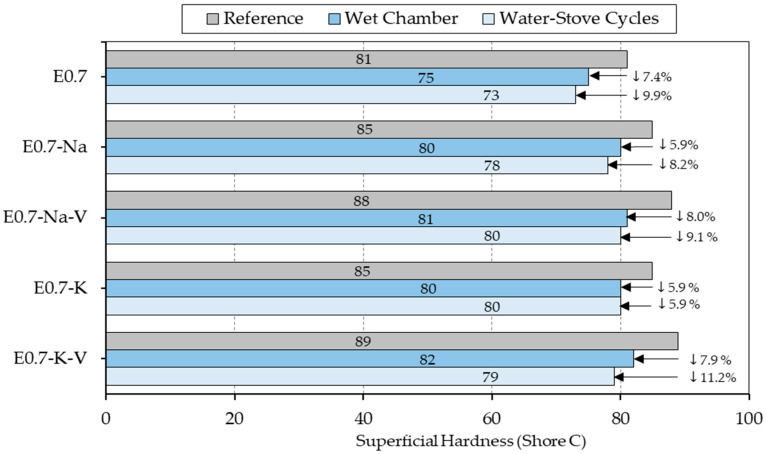
Shore C hardness test results compared.

**Figure 6 materials-16-00872-f006:**
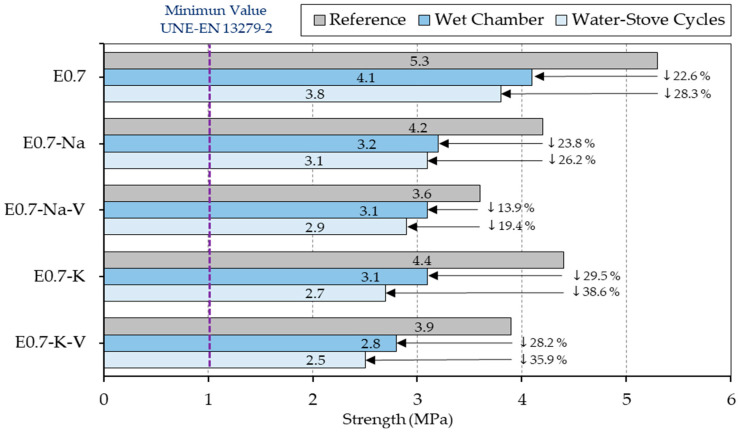
Flexural strength test results compared.

**Figure 7 materials-16-00872-f007:**
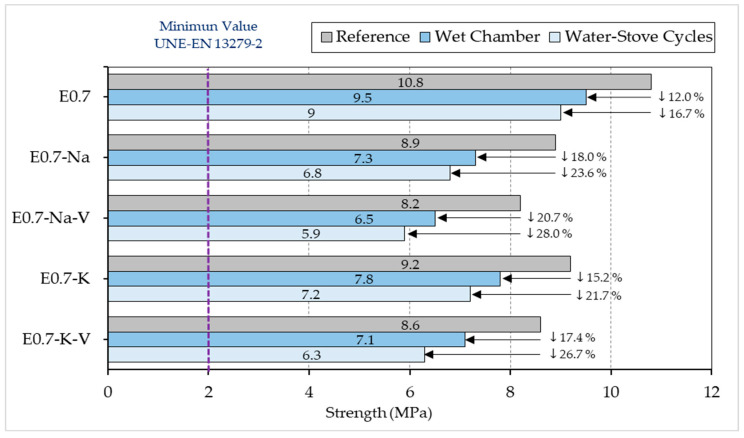
Compressive strength results compared (MPa).

**Figure 8 materials-16-00872-f008:**
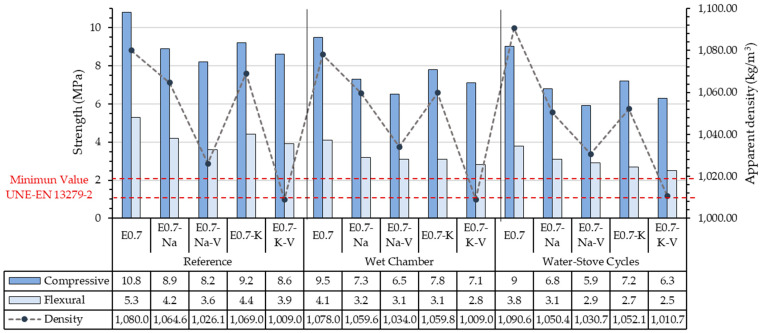
Results evaluation and comparison.

**Figure 9 materials-16-00872-f009:**
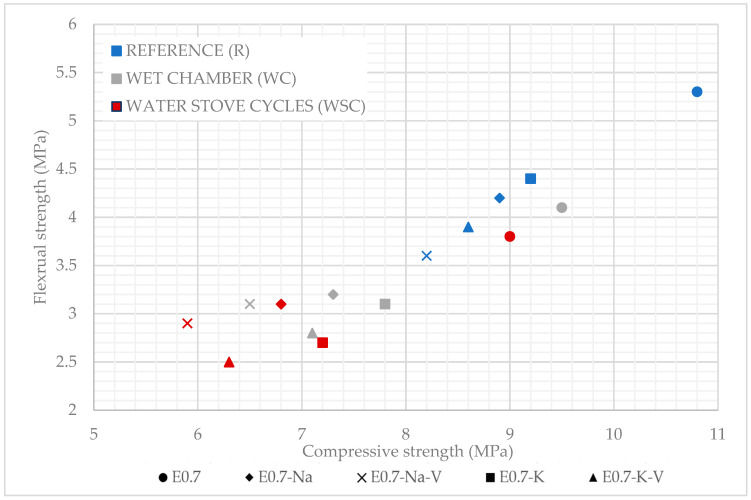
All mechanical performance results.

**Table 1 materials-16-00872-t001:** Super absorbent polymer (SAP) main characteristics [26,34].

	Appearance	Particle Size (µm)	Apparent Density (g/mL)	Humidity (%)	pH	Ignitability
(C_3_H_3_NaO_2_)_n_	Granulated	150–850	0.5–0.75	≤5	6.0 ± 0.5	Not flammable
(C_3_H_3_KO_2_)_n_	Crystals	1000–2000	0.56	≤5	6.8	Not flammable

**Table 2 materials-16-00872-t002:** Vermiculite properties.

Characteristics	Value
Appearance	Granulated
Particle size (mm)	1.4
Apparent density (kg/m^3^)	120
pH	7
Electrical conductivity (ms/m)	10
Ignitability	Not flammable

**Table 3 materials-16-00872-t003:** Dosages used and composition. Weight in grams (g).

Name	Plaster	Water	(C_3_H_3_NaO_2_)_n_	(C_3_H_3_KO_2_)_n_	Vermiculite
E0.7	1000	700	-	-	-
E0.7-Na	1000	700	15	-	
E0.7-Na-V	1000	700	15	-	30
E0.7-K	1000	700	-	15	
E0.7-K-V	1000	700	-	15	30

**Table 4 materials-16-00872-t004:** Samples elaborated and their uses.

Series	Dimensions	Tests
SERIES I	4 × 4 × 16 cm^3^	- Shore C hardness
		- Flexural strength - Compressive strength
SERIES II	Ø14 cm	- Water retention
SERIES III	4 × 4 × 16 cm^3^	- Capillarity water absorption
SERIES IV	4 × 4 × 16 cm^3^	- Total water absorption
		➢ Shore C hardness
		➢ Flexural strength
		➢ Compressive strength
SERIES V		- Water-stove cycles
		➢ Shore C hardness
		➢ Flexural strength
		➢ Compressive strength
SERIES VI		- Wet chamber
		➢ Shore C hardness
		➢ Flexural strength
		➢ Compressive strength
SERIES VII	Ø16.5 cm h: 2 cm	- Water vapour permeability

**Table 5 materials-16-00872-t005:** Water retention test results.

Type	Water Retention (%)
E0.7	61.77%
E0.7-Na	80.33%
E0.7-Na-V	70.19%
E0.7-K	84.20%
E0.7-K-V	73.00%

**Table 6 materials-16-00872-t006:** Water retention from water vapour permeability.

Specimen	Pre-Test Weight (g)	Post-Test Weight (g)	Difference (%)
E0.7	220.6	221.6	0.5%
E0.7-Na	282	283.7	0.6%
E0.7-Na-V	325.7	329.1	1.0%
E0.7-K	231.9	233.1	0.5%
E0.7-K-V	242.5	244.7	0.9%

**Table 7 materials-16-00872-t007:** Wet chamber test results weight variation.

Specimen	Weight (Dry) (g)	Weight after Test (g)	Δ(W)
E0.7	270.0	273.1	1.15%
E0.7-Na	271.3	276.4	1.88%
E0.7-Na-V	269.0	273.2	1.56%
E0.7-K	271.3	275.8	1.66%
E0.7-K-V	258.6	262.3	1.43%

**Table 8 materials-16-00872-t008:** Water–stove cycles test weight variation results.

Specimen	Initial Weight (Dry)	Weight C1 (Wet)	Δ(W) *	Weight C1 (Dry)	Δ(W) *	Weight C2 (Wet)	Δ(W) *	Weight C2 (Dry)	Δ(W) *
E0.7	273	360	32%	271.9	−0.40%	359.5	32%	272.4	−0.22%
E0.7-Na	274.2	379.4	38%	272.3	−0.69%	377.2	39%	272.8	−0.51%
E0.7-Na-V	272	370.6	36%	269.8	−0.81%	368.8	37%	270.2	−0.66%
E0.7-K	273.4	378.6	38%	271.2	−0.80%	375.1	38%	274.4	0.37%
E0.7-K-V	259.5	352.7	36%	258.8	−0.27%	350.1	35%	259.8	0.12%

* Weight compared to initial weight (dry). Weight in grams (g).

## Data Availability

Not applicable.

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
