# Peer review of "Characterization and under Water Action Behaviour of a New Plaster-Based Lightened Composites for Precast"

_materials, 2023, doi:10.3390/ma16020872_

Round 1

Reviewer 1 Report

This paper investigated the effect of the addition of sodium, potassium, and vermiculite to a new plaster-based lightened composite. A series of tests, including thermal test, mechanical test, and water action related tests, have been conducted on a series of samples. The tests were designed properly, and the results were sound and clear. The results clearly showed the property changes with the additions and will provide suggestions to applications. I recommend to publish it after making the following revisions:

1. Line 98, Line140, the titles seem to be written in another language, please change it to English, or provide necessary reasons.

2. Table 2 should be promoted to a location before section 2.2 for better reading experience.

3. Line 160. "In order to carry out...." This is hard to read. It sounds like "in order to do the tests, we design the tests". A sentence describing the goal of the tests should be provided. It should be like "In order to understand/investigate...., XXX tests will be conducted".

4. Line 312 to Line 315. The author claimed that the results are due to the water absorption capacity of the polymers, and results in a clogging of the pores. Is there any references or evidences (like images)?

5. It is suggested to provide a definition of the flexural strength. Is that the maximum stress generated in the sample?

6. Line 469-488. The author lists the findings in the conclusion part, it is recommended to rephrase them into one coherent paragraph.

7. Line 489-493. The description of no existing standard tests are not conclusion. It is recommended to promote to the introduction part, or the test methods part, or delete it.

Author Response

This paper investigated the effect of the addition of sodium, potassium, and vermiculite to a new plaster-based lightened composite. A series of tests, including thermal test, mechanical test, and water action related tests, have been conducted on a series of samples. The tests were designed properly, and the results were sound and clear. The results clearly showed the property changes with the additions and will provide suggestions to applications. I recommend to publish it after making the following revisions:

The authors would like to thank the reviewer for their evaluation, which has been very helpful in improving the quality of the work presented. All proposed changes have been made.

  1. Line 98, Line140, the titles seem to be written in another language, please change it to English, or provide necessary reasons.

Checked and changed.

  1. Table 2 should be promoted to a location before section 2.2 for better reading experience.

relocated, thank you.

  1. Line 160. "In order to carry out...." This is hard to read. It sounds like "in order to do the tests, we design the tests". A sentence describing the goal of the tests should be provided. It should be like "In order to understand/investigate...., XXX tests will be conducted".

The expression has been changed based on the reviewer’s comment.

“To develop this study, tests described below were conducted. With this, 7 series of three specimens of each chosen dosage have been prepared”

  1. Line 312 to Line 315. The author claimed that the results are due to the water absorption capacity of the polymers, and results in a clogging of the pores. Is there any references or evidences (like images)?

There are no images of this situation. As said SAP section water absorption increases 300 times its volume. With this, when water is passing through the sample, SAP is retaining water. Thus, when it reaches its maximum, water can not cross the sample. figure below may clarify this explanation

  1. It is suggested to provide a definition of the flexural strength. Is that the maximum stress generated in the sample?

It is the maximum load that the sample can withstand before breaking, applying this load in the direction perpendicular to its longitudinal axis. This is shown in Figure 2 (b). (Line 262)

  1. Line 469-488. The author lists the findings in the conclusion part, it is recommended to rephrase them into one coherent paragraph.

The autors thanks reviewer’s recommendation, but this point conflicts with other reviewer’s suggestions as well as other author’s publication in the same journal. Authors think this is a clearer way to express this research main points.

  1. Line 489-493. The description of no existing standard tests is not conclusion. It is recommended to promote to the introduction part, or the test methods part, or delete it.

Information has been deleted based on the reviewer’s comment

Reviewer 2 Report

The authors may consider to revise their abstract. If some quantity improvement compared to current materials can be addressed, it will show a clear statement to readers.

There is a 2 after the sentence. Please check what is the meaning on Line 95.

Author Response

The authors may consider to revise their abstract. If some quantity improvement compared to current materials can be addressed, it will show a clear statement to readers.

The authors would like to thank the reviewer for their evaluation, which has been very helpful in improving the quality of the work presented. All proposed changes have been made. Abstract suggestions have been added according to reviewer’s comment.

There is a 2 after the sentence. Please check what is the meaning on Line 95.

Mistake has been corrected based on reviewer’s comment

Reviewer 3 Report

The article concerns building materials made with polymers. The rules of writing scientific papers are it. The research results were discussed in the context of the research results of other scientists. They were summed up with correctly drawn conclusions.

The title of section 2.2 is written in Spanish.

Author Response

The article concerns building materials made with polymers. The rules of writing scientific papers are it. The research results were discussed in the context of the research results of other scientists. They were summed up with correctly drawn conclusions.

The authors would like to thank the reviewer for their evaluation, which has been very helpful in improving the quality of the work presented. All proposed changes have been made.

The title of section 2.2 is written in Spanish.

Thank you for the observation. Tittle has been checked and changed into English.

Reviewer 4 Report

The paper includes new data that are of scientific importance. The paper corresponds to the topics of scientific journal, it can be accepted for publication. Paper needs minor corrections.

1. The title of the article is too long, which is unacceptable. The authors only study the behavior of new lightweight gypsum-based composites under the action of water. This is confirmed by the authors themselves: "the aim is to check composites water reaction studied..." (lines 180 - 181) and "this research, water behaviour of a new type of lightened plaster composites has been studied" (line 461). The authors are invited to make the title shorter in order to reflect the content of the article and attract the attention of readers.

2. The abstract is representative of the content, but has no goal, scientific novelty. Keywords cannot duplicate the words and(or) phrases from the title of the paper.

3. Group references [7]-[12] (line 40) etc are not allowed. Section 2.2 - not English (line 140).

4. There is no processing of the experiment: what are the deviations, mathematical expectation and other parameters. It is necessary to specify the experiment processing parameters, such as mathematical expectation, deviations and etc. On Figures 3, 4 etc and some tables something similar is given without quantification.

5. Figure 9 (although the authors for some reason indicate section (line 441)) presents dispersion results in order to establish a selection of the samples with the best and worst behaviour under water action. But the authors are only talking about decrease in their mechanical properties is observed when subjected to accelerated water exposure cycles. The dispersion of a quantity is a measure of the spread of the values of a random variable relative to its mathematical expectation. It is usually measured as a percentage of the expected value.

6. Discussion section needs the slightly revision for the scientific standard:

 – what can be considered the advantages of this study compared to analogues (and it is mandatory to indicate alternative methods as a basis for comparison)

 – what are the shortcomings of the study

 – what could be the development of this research and why exactly in this.

Author Response

The paper includes new data that are of scientific importance. The paper corresponds to the topics of scientific journal, it can be accepted for publication. Paper needs minor corrections.

The authors would like to thank the reviewer for their evaluation, which has been very helpful in improving the quality of the work presented. All proposed changes have been made.

  1. The title of the article is too long, which is unacceptable. The authors only study the behavior of new lightweight gypsum-based composites under the action of water. This is confirmed by the authors themselves: "the aim is to check composites water reaction studied..." (lines 180 - 181) and "this research, water behaviour of a new type of lightened plaster composites has been studied" (line 461). The authors are invited to make the title shorter to reflect the content of the article and attract the attention of readers.

This article has water behaviour as the main goal, but mechanical and physical characterization test are also performed as a fully new material created by the authors. We think no to show this on the title could lead to confusions and it would not be accurate. Despite this, tittle has been rewritten according with reviewer’s suggestion.

  1. The abstract is representative of the content, but has no goal, scientific novelty. Keywords cannot duplicate the words and(or) phrases from the title of the paper.

Keywords have been changed based on reviewer’s suggestions.

  1. Group references [7]-[12] (line 40) etc are not allowed. Section 2.2 - not English (line 140).

Group references and section 2.2 have been changed based on reviewer’s comment.

  1. There is no processing of the experiment: what are the deviations, mathematical expectation and other parameters. It is necessary to specify the experiment processing parameters, such as mathematical expectation, deviations and etc. On Figures 3, 4 etc and some tables something similar is given without quantification.

This is a very interesting essay that is intended to be considered in future research. As the reviewer indicates, a statistical comparison and a study of deviation could give some extra information about a behavior prediction. Thus, in this study we focused on this material’s  mechanical and water action behavior, However, we want to thank the reviewer for his accurate comment that will be considered in future lines of research.The results are from the average of 3 samples and a statistical treatment would not be meaningful.

  1. Figure 9 (although the authors for some reason indicate section (line 441)) presents dispersion results in order to establish a selection of the samples with the best and worst behaviour under water action. But the authors are only talking about decrease in their mechanical properties is observed when subjected to accelerated water exposure cycles. The dispersion of a quantity is a measure of the spread of the values of a random variable relative to its mathematical expectation. It is usually measured as a percentage of the expected value.

Thank you for the observation. In this case, the mistake could be this figure’s name. The aim of this figure is to contextualize all the results obtained together to see which one performed better. As a result of this accurate reviewer’s comment, this figure has been renamed.

  1. Discussion section needs the slightly revision for the scientific standard:

 – what can be considered the advantages of this study compared to analogues (and it is mandatory to indicate alternative methods as a basis for comparison)

 – what are the shortcomings of the study

 – what could be the development of this research and why exactly in this.

Following reviewer’s comments, discussion section has been revised and some content added.